# Effect of Unloading Condition on the Healing Process and Effectiveness of Platelet Rich Plasma as a Countermeasure: Study on In Vivo and In Vitro Wound Healing Models

**DOI:** 10.3390/ijms21020407

**Published:** 2020-01-09

**Authors:** Francesca Cialdai, Alessandra Colciago, Desiré Pantalone, Angela Maria Rizzo, Stefania Zava, Lucia Morbidelli, Fabio Celotti, Daniele Bani, Monica Monici

**Affiliations:** 1ASA campus Joint Laboratory, ASA Res. Div., Department of Experimental and Clinical Biomedical Sciences “Mario Serio”, University of Florence, 50139 Florence, Italy; francesca.cialdai@unifi.it; 2Department of Pharmacological and Biomolecular Sciences, University of Milan, 20133 Milan, Italy; alessandra.colciago@unimi.it (A.C.); angelamaria.rizzo@unimi.it (A.M.R.); stefania.zava@unimi.it (S.Z.); fabio.celotti@unimi.it (F.C.); 3Unit of Surgery and Trauma Care, Department of Clinical and Experimental Medicine, University of Florence, 50134 Florence, Italy; desire.pantalone@unifi.it; 4Department of Life Sciences, University of Siena, 53100 Siena, Italy; lucia.morbidelli@unisi.it; 5Research Unit of Histology & Embryology, Department of Experimental and Clinical Medicine, University of Florence, 50139 Florence, Italy; daniele.bani@unifi.it

**Keywords:** wound healing, *Hirudo medicinalis*, platelet rich plasma, microgravity

## Abstract

Wound healing is a very complex process that allows organisms to survive injuries. It is strictly regulated by a number of biochemical and physical factors, mechanical forces included. Studying wound healing in space is interesting for two main reasons: (i) defining tools, procedures, and protocols to manage serious wounds and burns eventually occurring in future long-lasting space exploration missions, without the possibility of timely medical evacuation to Earth; (ii) understanding the role of gravity and mechanical factors in the healing process and scarring, thus contributing to unravelling the mechanisms underlying the switching between perfect regeneration and imperfect repair with scarring. In the study presented here, a new in vivo sutured wound healing model in the leech (*Hirudo medicinalis*) has been used to evaluate the effect of unloading conditions on the healing process and the effectiveness of platelet rich plasma (PRP) as a countermeasure. The results reveal that microgravity caused a healing delay and structural alterations in the repair tissue, which were prevented by PRP treatment. Moreover, investigating the effects of microgravity and PRP on an in vitro wound healing model, it was found that PRP is able to counteract the microgravity-induced impairment in fibroblast migration to the wound site. This could be one of the mechanisms underlying the effectiveness of PRP in preventing healing impairment in unloading conditions.

## 1. Introduction

Wound healing is the process that makes organisms resilient to injuries, allowing survival. Being a process of fundamental importance for life, it has been conserved throughout evolution. Wound healing is classically divided into three phases: inflammation, proliferation, and remodeling. Indeed, it consists of a very complex series of events and mechanisms, which partially overlap both spatially and temporally. After an injury, the blood clotting process starts immediately. Clot formation requires interactions among endothelial cells, platelets, and coagulation factors. Trapped cells and platelets within the clot trigger an inflammatory response by the release of vasodilators and chemoattractants and activation of the complement cascade [1]. The recruited neutrophils and macrophages produce pro-inflammatory cytokines and growth factors, resulting in migration and activation of (myo)fibroblasts, endothelial, and epithelial cells in the wound site, which are responsible for extracellular matrix deposition, neoangiogenesis, and re-epithelialization, respectively. Clot and debris are then removed by macrophages and extracellular hydrolases—matrix metalloproteases (MMPs), elastase, and plasmin protease—and the repair proceeds with wound contraction, extracellular matrix (ECM) secretion, and remodeling due to myofibroblasts and secreted metalloproteases.

Many factors can impair wound healing. A crucial factor is tissue hypoxia, which may be caused by primary vascular diseases, metabolic diseases such as diabetes, local and systemic infections, malnutrition, and persistent local pressure. In skin ulcers, the persistence of the inflammatory phase can lead to high protease activity, with degradation of growth factors and other molecular stimuli involved in tissue repair. Growth factors may be trapped by extracellular matrix molecules or degraded by proteases. Imbalance between extracellular hydrolases and their inhibitors results in abnormal ECM degradation [1,2]. Other alterations of the repair mechanisms, such as the persistence of stromal activation, can cause fibrotic scars and keloids. Moreover, pathogenic microorganisms can infect the wound (contaminated wounds and surgical site infections), stimulating a further immune response, inflammation, and tissue damage, as well as slowing the healing process and promoting abnormal tissue remodeling.

Skin wounds and compromised wound healing are major public health concerns. Complex and lengthy treatments cause an increasing burden on healthcare expenses. Even in uncomplicated cases, burns, chronic, and complicated wounds can require surgery and extended hospitalization periods. In the United States (US) alone, millions of patients need treatments for chronic wounds and an estimated US$ 50 billion is spent annually (data published by ‘Wound Care Awareness Week-2017’, including direct and indirect costs). More worryingly, the burden is growing year by year mainly owing to the increasing prevalence of obesity and diabetes. This problem is of similar importance in Europe; in the United Kingdom (UK), about 200,000 patients have a chronic wound [3], and in the European Community, the prevalence is expected to be over 1.5 million people [4,5].

Various medical approaches and therapeutic interventions can affect the different processes involved in wound healing. Topical application of growth factors and protease inhibitors, incision priming with platelet derived growth factor (PDGF) or interleukin-1 (IL-1), electric and magnetic field stimulation, exposure to laser radiation, negative pressure therapy, use of prosthetic materials, and gene and stem cell therapies have all been exploited to improve healing [6,7,8,9,10,11]. When a suture is needed, proper suture techniques and materials can strongly affect healing time and scar quality. In fact, the suture facilitates wound closure by obliteration of dead space, distribution of tension along deep suture lines, and maintenance of tensile strength in the injured tissue.

Among the therapeutic interventions, the application of platelet rich plasma (PRP) is one of the most widely used. PRP is a mixture of growth factors and cytokines, released by platelet alpha-granules degranulation. It is obtained from total blood by centrifugation and filtering; the fraction enriched in platelets is then activated using batroxobine or thrombin, thus degranulating [12]. PRP use is a simple and cheap method to promote tissue regeneration using autologous growth factors. When a tissue is damaged, the rapid initiation of the repairing phase is crucial to assure regeneration; the release of growth factors and cytokines stored in platelet alpha-granules, occurring within a few minutes of the damaging insult, represents a rapid support to tissue healing. For regeneration to occur, migration, proliferation, and differentiation of various cell types such as fibroblasts, endothelial cells, and mesenchymal stem cells are needed [12,13]. In particular, PRP efficacy in tissue regeneration is mainly thanks to its effects on the migration, proliferation, and biosynthetic activity of dermal fibroblasts, thus promoting their differentiation into myofibroblasts [14].

PRP has been used in different clinical conditions since the 1990s, starting from its first use as grafting for dental implants [15], and following with other more recent applications in diabetic ulcers, burns, and tissue recovery after surgery [12,16,17]. The interest in PRP’s clinical uses is still increasing, as demonstrated by a number of registered clinical trials where PRP is tested alone or in combination with scaffold materials or stem cells [18]. Nonetheless, the molecular basis underlying the use of PRP has not yet been fully elucidated. The management of serious wounds—for example, those resulting from trauma or emergency surgery—during space missions is a very challenging issue. Critical factors are, on one side, the limited availability of diagnostic and therapeutic tools, as well as the lack of a specialized medical staff on board space vehicles; and on the other side, the microgravity (*µg*)-induced alterations of human physiology, which can affect patient conditions and evolution of the healing process. Moreover, the microbial load and microorganism behavior onboard space vehicles should be considered to prevent/minimize the risk of wound infection and, in case an infection occurs, to have adequate countermeasures.

In current missions, the management of traumatic and surgical emergencies consists of patient stabilization and rapid return to Earth. In future interplanetary missions, timely medical evacuation to Earth would not be feasible and the guide of crew actions remotely would be useless because of the communication lag. Therefore, studies on the behavior and healing of sutured/non-sutured wounds in space environment are needed to understand possible problems and define adequate countermeasures [19]. The literature on wound healing in weightlessness is relatively poor. Studies on animal models in unloading conditions have given controversial results and no definite conclusions [20,21,22,23,24,25].

In vitro studies on immune cells, fibroblasts, endothelial, and epithelial cells cultured both in real and modeled *µg* conditions show alterations of functions involved in wound healing, such as phagocytosis, adhesion/migration, apoptosis, proliferation, intercellular cross-talking, production of inflammatory mediators, ECM molecules, growth factors, and so on [26,27,28,29,30,31,32].

In astronauts, deficient immune function, signs of chronic inflammation, metabolic alterations, and skin atrophy have been observed [33,34,35], and could affect the efficiency of tissue repair mechanisms.

In order to expand the knowledge on the behavior and healing of wounds and sutures in space environment, suitable models are needed to perform experiments onboard space vehicles as well as in modeled *µg* and hypergravity conditions on ground.

Recently, we developed an in vivo model of sutured and not sutured wound healing in weightlessness based on the use of leeches (*Hirudo medicinalis*), which is an adaptation of a widely accepted model for the study of basic tissue repair events in normal gravity (1× *g*) [36]. The leech is considered a reliable model because, despite being an invertebrate, it displays a wound healing process characterized by the same sequence of events (i.e., fibroplasia, angiogenesis, and remodeling) observed in vertebrates in response to injury. Indeed, the process involves the same cellular mechanisms and also the same types of molecules to regulate cell behavior in the different phases of healing [37]. Moreover, the leech is a particularly suitable model for experiments in space flight and in *µg*-modeling facilities owing to its small size and very long resistance to fasting [38].

The present paper describes the results of a study on the healing of sutured wounds in leeches exposed to weightlessness, modeled by a random positioning machine (RPM). Because, in weightlessness, the healing resulted delayed, the effectiveness of platelet rich plasma (PRP) in promoting repair mechanisms in µg was tested. Finally, PRP treatment was applied to an in vitro wound healing model, consisting of a scratch assay on fibroblast monolayers exposed or not to modeled *µg*. The effect obtained in vitro was analyzed and compared to that observed in vivo, with the aim to shed some light on the molecular and cellular mechanisms underlying PRP action in *µg* conditions.

## 2. Results

### 2.1. Effect of Modeled µg on In Vivo Model of Sutured Wound Healing

#### 2.1.1. Histological Analysis

To evaluate the effect of *µg* on the healing progress, the morphological features at the wound site were analyzed by histological techniques. In hematoxylin–eosin stained cross sections from animals exposed to modeled *µg*, light microscopical analysis of the tissues at the wound site (dorsal skin) showed a delayed healing in comparison with 1× *g* controls. At any time of exposure to *µg*, that is, 48 h, 96 h (Figure 1), and 144 h (Figure 4), the wounds appeared only partially covered by a loose epithelium characterized by poorly adherent cells, while control samples showed signs of a more organized re-epithelialization. The healing impairment was particularly evident after 144 h exposure; as shown in Figure 4, despite the considerable time elapsed from surgery, the dorsal muscle tissue was still adjacent to the wound margins, likely owing to the very scarce *de novo* formation of reparative connective tissue in the leeches longer exposed to *µg*.

#### 2.1.2. Analysis of Collagen Fibres

Collagens are very important in the healing progression. They are synthesized by fibroblasts and, assembling into fibres, constitute the basis for ECM formation within the wound and confer integrity and strength to the regenerating tissue. Therefore, the collagen fibre density at the wound site was evaluated in animals exposed to modeled *µg* and 1× *g* controls [39].

Light microscopical analysis and morphometric quantitation of picrosirius red-stained collagen fibres in the connective tissue close and beneath the surgical wounds from the dorsal skin of leeches exposed to modeled *µg* for 48 h, 96 h (Figure 2), and 144 h (Figure 5) compared with the relevant 1× *g* controls showed a reduction of the collagen meshwork, which reached statistical significance (*p* < 0.05) after 96 h and 144 h. Visually, in the animals exposed to *µg*, the collagen fibres appeared to be smaller and sparser than those in the 1× *g* controls. Particularly, after 96 h and 144 h, a substantial part of the stained collagen was not arranged in well-recognizable fibres, but rather formed irregular clumps.

#### 2.1.3. Analysis of Elastic Fibres

Elastin fibre density was analyzed in the wounded tissues of leeches exposed and not exposed to µg because elastin has an important structural role in the skin, imparting recoil and resistance. Elastin signaling in tissue repair is still underexplored, but it fits with the hallmarks of fetal scarless wound healing. In wounds of adult organisms, the appearance of elastin is delayed and the elastic fibre network is disorganized [40]. The effect of *µg* on elastin fibre formation is unknown.

Light microscopical analysis and morphometric quantitation of paraldehyde fuchsin-stained elastic fibres in the connective tissue close and beneath the surgical wounds from the dorsal skin of leeches exposed to modeled *µg* for 48 h, 96 h (Figure 3), and 144 h (Figure 6) showed no significant differences in the elastic meshwork compared with the relevant 1× *g* controls. Visually, in the animals subjected to *µg*, the elastic fibres appeared slightly smaller and sparser than those in the 1× *g* controls, but the measured differences did not reach statistical significance. Both in 1× *g* controls and in the animals exposed to *µg* conditions, the content in elastic fibres significantly increased with time (to note that the wounds were inflicted immediately before the exposure to *µg*; therefore, the exposure time coincided with the time elapsed from the wounding).

### 2.2. Effect of Modeled µg and PRP on In Vivo Model of Sutured Wound Healing

In this paper, PRP was evaluated as a possible countermeasure against the wound healing impairment in unloading conditions. Therefore, the effects of PRP treatment on morphology, collagen, and elastic fibre content at the wound site were analyzed in animals exposed/not exposed to *µg*.

#### 2.2.1. Morphological Analysis

In both controls (1× *g*) and *µg*-exposed animals (144 h), PRP promoted wound healing by narrowing the surgical wound and enhancing re-epithelialization (Figure 4); in all the PRP-treated animals, exposed or not to modeled *µg*, the dorsal lesion was covered by a thick regenerative epithelium.

#### 2.2.2. Analysis of Collagen Fibres

Light microscopical analysis and morphometric quantitation of picrosirius red-stained collagen fibres in the peri-lesional connective tissue from the leeches exposed for 144 h to *µg* showed an appreciable, significant reduction of the collagen meshwork as compared with the 1× *g* controls. The treatment with PRP prevented the decrease in collagen meshwork density induced by exposure to *µg* (Figure 5).

#### 2.2.3. Analysis of Elastic Fibres

Light microscopical analysis and morphometric quantitation of paraldehyde fuchsin-stained elastic fibres in the connective tissue close to the surgical wounds on the back of leeches exposed to *µg* for 144 h (Figure 6) showed no significant changes of the elastic meshwork compared with the relevant controls (1× *g*). The treatment with PRP did not result in appreciable, significant changes in both control and *µg*-exposed animals.

### 2.3. Effect of Modeled µg and PRP on In Vitro Model of Wound Healing

#### 2.3.1. Morphological Changes

As fibroblasts play a pivotal role in tissue repair, alterations in their activity could strongly compromise healing. On the other hand, they could be a target for countermeasures. Therefore, part of this study focused on the fibroblast behavior, using fibroblast cultures and an in vitro model of wound healing; that is, the scratch assay. Moreover, the effect of PRP on fibroblasts exposed/not exposed to *µg* was evaluated.

After 72 h exposure to modeled *µg*, changes were observed in cell culture appearance and cell morphology; the flat monolayer and flat cell shape typical of 1× *g* control samples (Figure 7a) was partially substituted by three-dimensional (3D) aggregates (Figure 7b) in fibroblast cultures exposed to *µg*. PRP treatment during the exposure to *µg* prevented the formation of 3D aggregates (Figure 7d).

#### 2.3.2. Cell Count

The increase in fibroblast number at the wound site, which generally occurs in conjunction with the proliferative phase, is an early and fundamental event in wound healing; therefore, it was studied in control samples, in modeled *µg* and after treatment with PRP.

NIH-3T3 cell count was performed after 72 h exposure to modeled *µg* in the presence/absence of PRP (Figure 8). Fibroblast number significantly decreased in samples exposed to *µg* compared with the 1× *g* controls. The presence of PRP during the exposure to *µg* did not significantly counteract the effect induced by *µg* on the cell number.

#### 2.3.3. Cell Migration

Following injury, fibroblasts migrate into the wound from the surrounding tissues, being attracted by factors that are released by inflammatory cells and platelets. Fibroblast accumulation in the wound can be observed on the third day after injury. Once in the wound, they proliferate profusely and produce ECM. By the end of the first week, fibroblasts change to their myofibroblast phenotype, responsible for wound contraction. Therefore, the migration of fibroblasts to the wound is essential for the proper performance of subsequent events [39]. The ability of fibroblasts to migrate and close a wound was evaluated by an in vitro scratch assay, with the scratch being performed immediately before the exposure to modeled *µg* (72 h). Samples with and without PRP (Figure 9) were monitored. Photos acquired by light microscopy (20× magnification) during the scratch assay in controls (1× *g*) and *µg*-exposed samples (Figure 9A), as well as measurements of migration speed (Figure 9B), clearly showed that, during clinorotation (that is the exposure to modeled *µg* by RPM), the cell ability to migrate and close the wound was greatly impaired. The addition of PRP during clinorotation was effective in preventing the impairment of cell migration; PRP induced a statistically significant faster closure of the scratch in *µg*-exposed cell cultures compared with PRP-untreated *µg*-exposed cells. Similarly, when cell migration was assessed towards a chemoattractant (fetal calf serum (FCS) 1%) in a Boyden chamber assay (Figure 10) performed after exposure to *µg*, a statistically significant lower number of *µg*-exposed cells migrated towards the chemoattractant, compared with 1× *g* controls. Again, when PRP was present in the culture medium during the exposure to modeled *µg*, the impairment of chemo-responsivity in *µg*-exposed cells was partially reversed; PRP induced a statistically significant increase in the chemotactic response compared with PRP-untreated *µg* exposed cells (Figure 10).

#### 2.3.4. Gene Expression

α-smooth muscle actin (α-SMA) is a contractile protein. It is a marker of fibroblast-myofibroblast transdifferentiation; myofibroblast phenotype is characterized by the presence of thick α-SMA boundles and pseudopodia, which allow the attachment to fibronectin and collagen in the ECM. Wound contraction takes place as these cell extensions retract [39].

Vascular endothelial growth factor (VEGF) is a glycoprotein that shows unique effects on multiple components of the wound healing cascade, including angiogenesis, epithelialization, and collagen deposition [41]. VEGF is produced by many cell types involved in wound healing, including fibroblasts. VEGF is a powerful stimulator of endothelial cell migration, proliferation, and capillary tube formation. As angiogenesis maintains a critical role in wound healing, VEGF is regarded as a possible therapeutic factor to manage non-healing wounds [42].

Considering that α-SMA and VEGF regulate key events in wound healing, their expression was assessed by real time PCR on total RNA from in vitro models of wound healing (scratched fibroblast monolayers) maintained at 1× *g* (controls), treated/untreated with PRP, or exposed to modeled *µg* for 72 h, treated/untreated with PRP during the exposure. Data are expressed as 2^−∆∆*C*t^ versus 1× *g* controls (Figure 11).

The expressions of α-SMA (A) and VEGF (B) were strongly influenced by modeled *μg*. As shown in panel A, α-SMA expression was significantly reduced in fibroblasts exposed to modeled *μg*. The treatment with PRP induced a significant increase in α-SMA expression in 1× *g* controls, but it was ineffective in counteracting α-SMA decrease in *μg*-exposed samples. The weightlessness conditions induced a strong increase in VEGF expression, which was effectively prevented by adding PRP during the exposure to *μg*.

## 3. Discussion

Studying wound healing in space is interesting for two main reasons: (i) understanding the role of gravity and mechanical factors in the healing process and scarring, thus contributing to unraveling the mechanisms underlying the switch between full regeneration and scarring; (ii) defining tools, procedures, and protocols to manage serious wounds and burns eventually occurring in future long-lasting space exploration missions. However, owing to the constraints imposed by the use of devices to model *μg* or platforms allowing experiments in real *μg*, these studies need models with specific requirements. The leech is accepted as a reliable in vivo model to study wound healing; although it is a relatively simple organism, its tissue repair processes show a striking similarity with those of vertebrates. Moreover, the leech is considered an effectual model to test the action of pharmacological and non-pharmacological treatments [43].

We hypothesized that the leech could also be a reliable model to study wound healing in altered gravity conditions as well as for experiments to be performed in space. Indeed, the present results demonstrate that the sutured wound model proposed here is sensitive to *μg* and PRP treatment, tested as a countermeasure for *μg*-induced anomalies in the healing process.

In the animals wounded, sutured, and exposed to *μg* for times varying from 48 h to 144 h, healing was delayed in comparison with 1× *g* controls. Histology revealed wounds with newly formed connective and epithelial tissues, which appeared poorly organized. These structural alterations were further confirmed by the decrease in collagen fibre density in *μg*-exposed animals, while elastic fibres did not differ significantly from 1× *g* controls. It is well known that elastic fibres appear at a late stage of healing. Therefore, the relatively short exposure to *μg* (max 144 h), which, in the described experiments, was performed immediately after wounding, apparently affected far more early than late events.

So far, relatively few studies on wound healing have been conducted in animal models exposed to unloading conditions [20,21,23,24,25,44]. Most of them focused on healing of bone fractures and ligaments in rodent models; therefore, there is a lack of information on soft tissue repair and the comparison with our study is quite difficult. However, in full agreement with our results, most of the studies reported healing delay characterized by reduced ECM deposition, which consequently jeopardized repair mechanisms in connective tissues [45,46].

The addition of PRP to the medium during exposure of the wounded leeches to *μg* prevented both healing delay and alterations in tissue structure. On Earth, PRP is widely used to favour healing in wounds and ulcers. The results of this study demonstrate that PRP is also effective in promoting wound healing in unloading conditions. Therefore, in view of future long-lasting space exploration missions, PRP should be considered among the possible countermeasures to manage wound healing in space.

In a previous in vitro study [31], we observed that PRP treatment, performed in fibroblast cultures after exposure to *μg*, modeled by a rotating cell culture system (RCCS), was able to partially counteract *μg*-induced alterations in fibroblast functions involved in wound healing. As fibroblasts play a crucial role in wound healing, we hypothesized that the effect of PRP on fibroblasts could be one of the mechanisms underlying the PRP effectiveness in promoting wound healing in the *μg*-exposed leeches. To verify this hypothesis, in the present study, we also evaluated the response of fibroblasts to PRP, adding it to the culture medium during the exposure of the samples to *μg*. After 72 h exposure, we observed clear changes in cell morphology as well as in cell distribution, with formation of aggregates instead of an ordered monolayer. PRP treatment prevented the *μg*-induced formation of 3D aggregates. The three-dimensional growth observed after 72 h RPM exposure is a known effect of unloading conditions on different cell types [47,48,49]. During rotation, cells are not driven against a solid surface and do not grow across a solid–liquid interface, but rather tend to form 3D aggregates. This behavior is also associated to cytoskeletal disruption and reorganization [49]. PRP ability to counteract the *μg*-induced formation of 3D aggregates might be related to its cytoskeletal remodeling abilities [50].

In modeled *μg* conditions, cell count revealed a decrease in fibroblast number and PRP treatment was unable to counteract this effect, which could be because of cell death/apoptosis or the reduced proliferation rate. The *µg*-induced decrease in proliferation [47,49] as well as increase in apoptosis [28,29] have been described in different cell types. To our knowledge, an increase in apoptosis has never been described in fibroblasts cultured in real or modeled *µg*, and we did not observe necrosis and/or apoptotic bodies in fibroblast cultures (see Figure 7). Sun et al. [51] reported that *µg*-induced reduction in osteoblast proliferation is because of an arrest in cell cycle progression. In agreement with Sun, in experiments on fibroblasts cultured in *µg*, modeled by a rotating cell culture system (RCCS), we found a temporary arrest of the cell cycle, with repercussion on proliferation (data not yet published). On the basis of the above reports in the literature and our observations, it could be speculated that the decrease in fibroblast number was the result of a reduced proliferation rather than an increased apoptosis. However, considering that proliferation and apoptosis have key roles in healing, they need to be studied in depth in fibroblasts and other cells involved in tissue repair process.

In the in vitro wound healing models (scratched fibroblast monolayers) exposed to unloading conditions, the ability of fibroblasts to migrate was significantly decreased, as assessed by both scratch and Boyden chamber assays. Previous studies demonstrated an altered production of ECM molecules, such as fibronectin and collagen I, in fibroblasts and endothelial cells cultured in *μg* simulated by RPM [30]. A relationship could be speculated between inability to migrate and altered ECM production, which could be also a cause of the scarce *de novo* formation of reparative connective tissue observed in the leeches exposed to *μg* (Figure 4).

Moreover, in *μg*-exposed models, the expression of α-SMA, a key marker of fibroblast activation, decreased while the expression of VEGF increased as compared with the 1× *g* controls.

Cytoskeletal remodeling, formation of 3D-aggregates, and proliferation decrease might be tightly related to α-SMA expression modification. α-SMA is an actin isoform strictly related to myofibroblast activation [52], as its basal expression (absence of tissue injuries) is markedly increased during tissue repair. Indeed, the activated myofibroblasts can contract, merging wound edges and speeding up healing [53]. In Earth gravity conditions (1× *g*), PRP-induced α-SMA expression might represent one of the multiple mechanisms by which PRP facilitates tissue regeneration [54]. In unloading conditions, PRP was ineffective in preventing the decrease in α-SMA, suggesting a weaker or slower effect on myofibroblast activation. The present results largely confirm what we observed in a previous study on fibroblasts cultured for 72 h in RCCS [31]. A significant difference concerns the VEGF expression; following exposure to simulated μg, it increased in the present study, while it was found to be decreased in the previous one. This different behaviour could reflect the differences in the models used; an in vitro wound healing model (scratch on fibroblast monolayer) exposed to *μg* simulated by RPM and a culture of fibroblasts adherent on beads exposed to *μg* simulated by RCCS, respectively. It has been demonstrated that *μg* per se induces an increase in inflammatory signals [31]. Therefore, in the present study, the stimulus due to the scratch may have reinforced the inflammatory response, which would justify the increase in VEGF transcription. Moreover, it is known that VEGF transcription is induced by hypoxia [55]. In the internal core of 3D aggregates observed in this study following the exposure to *μg*, modeled by RPM, hypoxia was probably present and may have induced VEGF expression. The ability of PRP to prevent 3D aggregates formation might be related to the reduced VEGF expression obtained after PRP treatment.

During the exposure of the samples to unloading conditions, PRP treatment was effective in preserving the fibroblast ability to migrate, as demonstrated by the scratch and Boyden chamber assays. PRP, as a concentrated mixture of growth factors, is known for its ability to induce proliferation and migration in many cell types [55]. Thus, the experiments described here were done in low serum conditions to reduce the possible proliferative effect of PRP on healing closure velocity. Moreover, as stated above, PRP did not influence fibroblast proliferation either in 1× *g* control or in *µg*-exposed cells. PRP was also effective in preventing VEGF increase. On the contrary, PRP was ineffective in preventing the decrease in α-SMA, which would suggest a weaker or slower effect on fibroblast activation.

In conclusion, to the best of our knowledge, this is the first report about the use of an in vivo sutured wound healing model in the leech to study soft tissue wound repair in unloading conditions. The present results demonstrate that modeled *μg* delays the wound healing process both in vitro and in vivo, and that PRP treatment is effective in counteracting some of the *μg*-induced alterations in the healing process, both in vitro and in vivo. PRP action seems mostly the result of its effectiveness in preserving the ability of fibroblast to migrate at the wound site. The weaker/slower effect of PRP on fibroblast activation observed in *μg* conditions raises intriguing questions. Currently, the main hypothesis on the mechanisms that switch from full regeneration to scarring argues that those favoring faster, albeit qualitatively inferior, tissue repair more commonly occur because they offer an evolutionary advantage over those inducing a slower, albeit better, tissue regeneration [56,57]. Studies on wound healing and tissue regeneration in *μg* can help understand this important biological problem and pave the way for the development of therapeutic strategies to favor regeneration instead of scarring.

## 4. Materials and Methods

### 4.1. Animals and Surgical Treatment

Leeches (*Hirudo medicinalis*, Annelida, Hirudinea) measuring about 7.00 cm in length and 1.00 cm in diameter were purchased by Ricarimpex (Eysines, France). In 2004, this company received Food and Drug Administration (FDA) clearance to market leeches as medical devices to favor skin grafts healing, reattachment surgery, and the restoration of blood circulation by removing pooled blood.

Leeches were housed in aerated aquaria containing mineral water at a temperature of about 20 °C. The leeches were fed 15 days before the experiment. As leeches can last for more than one month without food, it was not necessary to feed them during the experiment.

Before surgical procedures, leeches were anesthetized by immersion in 10% ethanol (AppliChem GmbH, Darmstadt, Germany). A surgical lesion (length 10 mm, depth ~2 mm) was performed on the dorsal skin (at about the 14th metamere) of each leech with a 15° scalpel (Benefis srl, Genova, Italy). The incision was made, taking care to avoid the complete incision of the outer longitudinal muscle layer. Three stitches in a non-absorbable 4-0 polyamide monofilament suture with a 3/8 cutting needle (B. Braun Surgical, Rubi, Spain) were made to close the wounds.

### 4.2. Exposure to Modeled µg—Leeches

After the surgical procedure, the leeches were individually placed into T25 cell culture flasks (Corning Life Sciences, Tewksbury MA, USA) and randomly divided into separate experimental groups, as follows:

Group C—control animals maintained for 48 h, 96 h, or 144 h at 1× *g*;

Group *µg*—animals exposed for 48 h, 96 h, or 144 h to *µg* modeled by RPM;

Because, as expected, 144 h exposure better evidenced the effect of *µg* on the healing process, in these experiments, the effectiveness of PRP as a countermeasure was tested by including the following groups:

Group C + PRP—animals treated with PRP and maintained for 144 h at 1× *g*

Group *µg* + PRP—animals treated with PRP and exposed for 144 h to *µg* modeled by RPM.

For the exposure to modeled *µg*, the T25 flasks containing leeches were completely filled with mineral water in order to avoid air bubbles and shear stress. The exposure to *μg* coincided with the events occurring in the first week of the wound healing process.

In particular, the leech is an excellent model for studying fibroblast function and ECM production; a few hours after injury, fibroblasts in the surrounding tissues are stimulated to proliferate and then migrate to the wound site. Fibroblasts accumulate in the wound (third day), where they continue to proliferate and produce ECM and biochemical factors, such as VEGF, which regulate the activity of other cell populations. By the end of the first week, they change to their myofibroblast phenotype.

### 4.3. PRP Preparation

Platelet rich plasma was obtained from blood of healthy volunteers and prepared as previously described [58]. Briefly, blood was collected in citrate phosphate dextrose (Merck KGaA, Darmstadt, Germany) as anti-coagulant. The whole blood was centrifuged at 180× *g* for 20 min to separate platelets (upper layer) from red and white blood cells (lower layer). The upper layer was transferred into clean tubes and centrifuged at 580× *g* for 15 min; the platelet pellet was resuspended in a small volume of supernatant giving the final PRP fraction (platelet enrichment of about 4–5-fold, resulting in about 1.0–1.2 × 10^6^ platelets/μL). PRP was then activated with calcium gluconate/batroxobine (Pentapharm, Basel, Switzerland), giving an insoluble gel, where the activated platelets degranulate and release growth factors and cytokines. Platelet gel was centrifuged (1400× *g*, 10 min, room temperature) and the growth factors-enriched liquid phase was frozen at −20 °C till use (PRP). Under these storage conditions, the preparation remains effective for many months. For all the experiments described in this study, PRP was quickly thawed and added to mineral water (leeches) or culture medium (cells) at 1:1000 final dilution.

### 4.4. Random Positioning Machine Exposure

Simulation of *µg* was achieved by a random positioning machine (RPM, Dutch, Space, Leiden, The Netherlands). This is a three-axis clinostat, in which the weight vector is continuously reoriented as in traditional clinorotation, but with increased directional randomization. In the RPM, samples are fixed as close as possible to the center of the platform consisting of two frames rotating one inside the other, driven by separate motors. The rotation of each frame is random and autonomous under computer control. The *µg* conditions are modeled by averaging the gravity vector via the independent rotation of the two frames. The outer frame rotated perpendicular to the inner frame, which caused the samples to move randomly in three axes.

To simulate *µg*, the instrument was set in real random mode (random intervals, time, and rotation); the maximal angular velocity of rotation was set at 60°/s (1 rads/s). The samples contained in a T25 flask, filled with mineral water or medium (depending on experiments with leeches or cells), were fixed in the center of the inner frame reaching 10^−3^× *g* [59,60].

The RPM was accommodated in a temperature-controlled room, set at 20 °C for the exposure of leeches and 37 °C for the exposure of cells. Then, 1× *g* controls were placed on the fixed base of the RPM, facing the same vibrations and temperature as the rotating ones.

### 4.5. Histological and Morphometrical Analysis

After exposure to modeled *µg*, leeches from the different experimental groups were anesthetized by immersion in a solution of 10% ethanol (AppliChem GmbH, Darmstadt, Germany) in water and then sacrificed by immersion in 95% ethanol for 15 min. The mid part of the body with the surgical wound was dissected, fixed in 4% paraformaldehyde in phosphate buffer (Bio Optica Milano s.p.a., Milano, Italy) for 24 h, dehydrated in graded ethanol, and embedded in paraffin (Diapath S.p.A., Bergamo, Italy). Histological cross sections, 5 μm thick, were cut with a MR2 microtome (RMC Boeckeler, Tucson, AR, USA). Some sections were stained with hematoxylin and eosin (Bio Optica Milano s.p.a., Milano, Italy) for conventional histological observation. Other sections were stained with Picrosirius red (Sigma-Aldrich, St. Louis, MO, USA) for assessment of collagen fibres content or with paraldehyde fuchsin (Acros Organics, Geel, Belgium) for assessment of elastic fibre content in the tissues around and beneath the surgical wound. Stainings were made in a single session to minimize artifactual staining differences. Digital photomicrographs were taken with a Nikon Eclipse E200 light microscope with a ×40 objective, equipped with a Nikon DS Fi2 digital camera and NIS Elements image acquisition software (all from Nikon, Florence, Italy). Surface area measurements of the red-stained collagen fibers or the violet-stained elastic fibres were carried out using the ImageJ 1.33 image analysis program (National Institutes of Health, USA; http://rsb.info.nih.gov/ij) upon selection of four regions of interest (RoI) for each image and an appropriate threshold to only include the stained fibres. The regions of interest (RoI) used for collagen and elastic fibre morphometry were selected from the sub-epidermal and inter-muscular stroma flanking the margins and bed of the wound (Figure 12).

### 4.6. Cell Culture

Fibroblasts (NIH-3T3) were routinely cultured in Dulbecco’s modified Eagle’s medium (DMEM) supplemented with 100 μg/mL streptomycin, 100 U/mL penicillin, 2 mM glutamine, and 10% fetal bovine serum (FBS) (cell culture reagents from Euroclone S.p.A., Pero, Italy). Cells were incubated at 37 °C in humidified atmosphere containing 95% air and 5% CO_2_.

### 4.7. Exposure to Modeled µg—Cells

For the exposure to modeled *µg*, NIH-3T3 cells (4 × 10^4^) were seeded into T12.5 flasks (Corning Life Sciences, Tewksbury MA, USA). After 24 h, the flasks were completely filled with culture medium to avoid shear stress, in presence/absence of PRP, and then exposed to modeled *µg* for 72 h. Cell experimental groups were as follows:Group C—control samples, maintained for 72 h at 1× *g*;Group *µg*—samples exposed for 72 h to *µg* modeled by RPM;Group C + PRP—samples treated with PRP, maintained for 72 h at 1× *g*;Group *µg* + PRP—samples treated with PRP, exposed for 72 h to *µg* modeled by RPM.

### 4.8. Cell Count

At the end of the exposure to modeled *µg*, medium was removed and cells were washed with Ca^2+^- and Mg^2+^-containing PBS (pH 7.4) (Euroclone S.p.A., Pero, Italy). Cells were then detached by trypsinization (Trypsin 0.05%–EDTA 0.02%, Euroclone S.p.A., Pero, Italy) and counted with a Burker’s chamber under an inverted light microscope (Nikon, Amsterdam, The Netherlands). The same procedure was applied to 1× *g* control samples.

### 4.9. Migration Assays

#### 4.9.1. Scratch Test

Scratch assay is an in vitro model of wound healing used to study cell migration in response to injury. Cells of 8.5 × 10^5^ were seeded in T12.5 flasks and allowed to adhere. After the cells reached subconfluence, they were incubated for 24 h in DMEM + 0.5% FBS (Euroclone S.p.A., Pero, Italy) to reduce proliferation. Then, the culture medium was removed and a scratch was made on the monolayers with a plastic tip. Wounded monolayers were washed with Ca^2+^- and Mg^2+^-containing PBS (pH 7.4) to remove dead cells, and fresh complete medium, in presence/absence of PRP, was added. Experimental groups of samples were prepared as previously described. Wounded monolayers were observed and photographed immediately after the scratch (before being placed on RPM) (t0) and after 24 h and 48 h of exposure to modeled *µg*. A scanning microscope (Axiovert 200 Zeiss, Jena, Germany) and the MetaVue software (Molecular Devices, Sunnyvale, CA, USA) were used. Distances between cell fronts were measured with Image-ProPlus 6.0 (MediaCybernetics, Bethesda, MA, USA), considering at least six measurements from the top to the bottom. To capture images at 24 h and 48 h of exposure, RPM was stopped for a few minutes to allow samples displacement to the microscope and repositioning on the machine.

#### 4.9.2. Boyden Chamber Assay

The microchemotaxis assay was performed using a 48-well Boyden’s chamber according to manufacturer’s instructions (Neuroprobe, Cabin John, MD, USA). NIH-3T3 cells previously exposed to modeled *µg* or maintained at 1× *g* for 72 h, in presence/absence of PRP, were used for the assay (see experimental groups described above). At the end of the exposure, cells were immediately collected by trypsin, resuspended in DMEM + 0.1% bovine serum albumin (BSA, Merck KGaA, Darmstadt, Germany), and used to fill the upper compartment (3.5 × 10^4^ cells/well) of the chamber. FBS 1% was used as chemoattractant in the lower compartment of the chamber. Cells migrated through a polyvinylpyrrolidone-free polycarbonate porous membrane (8 μm pores, Biomap, Milano, Italy) pre-coated with gelatin (0.2 mg/mL in PBS, 5 days at 4 °C, Merck KGaA, Darmstadt, Germany).

After migration (overnight, 37 °C), cells adherent to the underside of the membrane were fixed by methanol (Merck KGaA, Darmstadt, Germany) and stained according to the Diff-Quik kit (Biomap, Milano, Italy). For quantitative analysis, cells were photographed using an optical microscope with a digital camera and counted using Image J software (National Institutes of Health, USA; http://rsb.info.nih.gov/ij). Three random objective fields were counted for each well and the mean number of migrating cells was calculated.

#### 4.9.3. Gene Expression

The expression of alpha-SMA and VEGF proteins was assessed by qRT-PCR on total RNA obtained from samples belonging to the four different experimental groups: NIH-3T3 cells cultured at 1× *g* (controls), treated/untreated with PRP, and NIH-3T3 cells exposed for 72 h to modeled *µg*, treated/untreated with PRP. Before exposure to *µg* and PRP treatment, a scratch was performed on the fibroblast monolayers to stimulate cell activation. The same procedure was performed on the related controls.

#### 4.9.4. RNA Extraction, Purification and Quantitation

Total RNA was extracted with RNeasy Mini kit (Qiagen, Hilden, Germany) according to the manufacturer’s instructions and then quantified with Nano-Drop2000 (Thermo Scientific, Waltham, MA, USA).

#### 4.9.5. Quantitative Real-Time PCR

Reverse transcription was performed on 1 microgram of total RNA from each sample according to the manufacturer’s protocol (iScript cDNA synthesis kit, BioRad, Segrate, Italy) using random primers. qPCR was performed in singleplex in CFX96 Touch™ Real-Time PCR Detection System (BioRad, Segrate, Italy) using SYBR Green dye (SsoAdvanced SYBR Green Supermix, Bio-Rad, Segrate, Italy) and specific sets of primers as follows:Alpha-SMA: 5′-CCCTGAAGAGCATCCGACAC-3′ and 5′ GCATAGCCCTCATAGATAGGCA-3′;VEGF: 5′-AAAACACAGACTCGCGTTGC-3′ and 5′-CTCCTAGGCCCCTCAGAAGT-3′;GAPDH: 5′-CCTGCGACTTCAACAGCAAC-3′ and 5′-TAGGGCCTCTCTTGCTCAGT-3′.


Data analysis was performed using the CFX Manager 2.0 software (Bio-Rad, Segrate, Italy). Each sample was analyzed in triplicate. Data were normalized for GAPDH Ct value. Relative mRNA levels were then calculated by the comparative Ct method (2^−ΔΔCt^) and data were expressed as fold induction versus control.

### 4.10. Statistical Analysis

Animal and cell experiments were carried out in triplicate.

Morphometrical data were represented as means ± SEM of the measurements of four individual animals, with at least five regions of interest (ROI, about 10,000 µm^2^ each), from the different experimental groups. Statistical comparison of differences between groups was carried out using one-way analysis of variance (ANOVA) followed by Student–Newman–Keuls multiple comparison test. A *p*-value ≤ 0.05 was considered significant. Calculations were done using GraphPad Prism 5.0 statistical program (GraphPad Software, San Diego, CA, USA).

The statistical analysis of migration assays was carried out with the Prism4 software for Macintosh (GraphPad Software, San Diego, CA, USA) and represented as means ± SD. Comparisons among the experimental groups were performed by Tukey’s multiple comparison test. A *p* value ≤ 0.05 was considered significant.

## Figures and Tables

**Figure 1 ijms-21-00407-f001:**
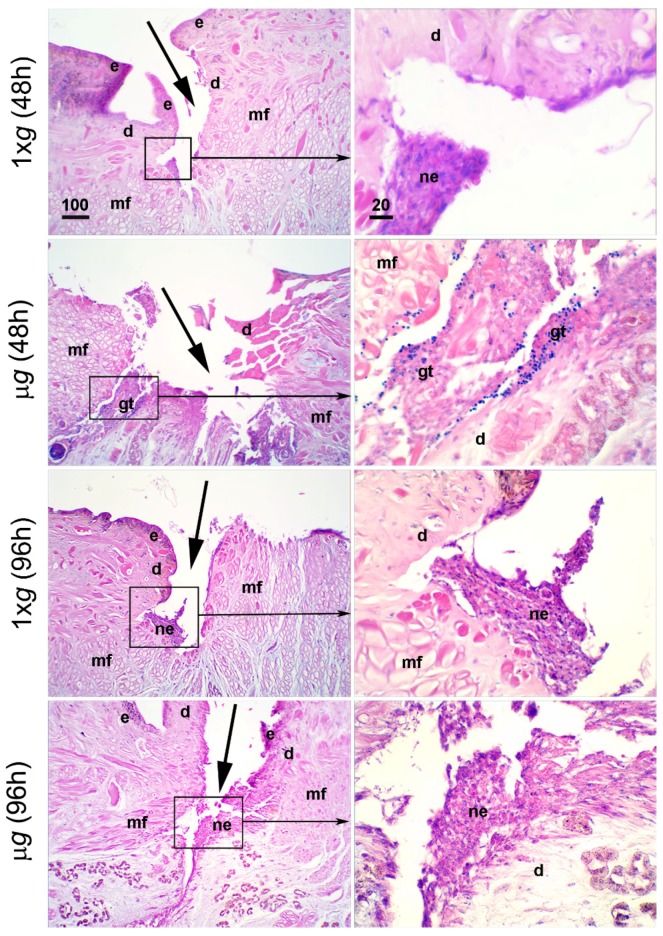
Effect of modeled *µg* on in vivo model of sutured wound healing (*Hirudo medicinalis*): state of the wound after 48 h and 96 h exposure. Hematoxylin–eosin stained cross sections. The panels show the state of the wounds on the back of the leeches (arrows) in the 1× *g* controls and in the animals exposed to modeled *µg* for 48 h or 96 h. Letterings are as follows: e, epidermis; d, dermis; mf, muscle fibers; ne, neo-epidermis; gt, granulation tissue. Calibration bars are in µm, as indicated.

**Figure 2 ijms-21-00407-f002:**
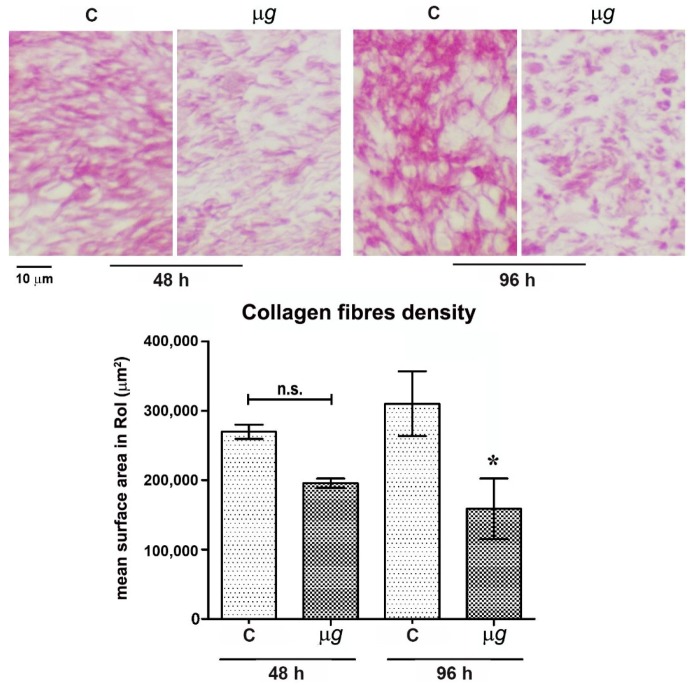
Effect of modeled *µg* on in vivo model of sutured wound healing (*Hirudo medicinalis*): collagen fibres content in the connective tissue at the wound site. Light microscopical features and morphometric quantitation of picrosirius red-stained collagen fibres at the wound site after 48 h and 96 h exposure to modeled *µg*. C = 1× *g* controls; * *p* < 0.05 vs. C-96 h (*n* = 3); n.s. = not significant.

**Figure 3 ijms-21-00407-f003:**
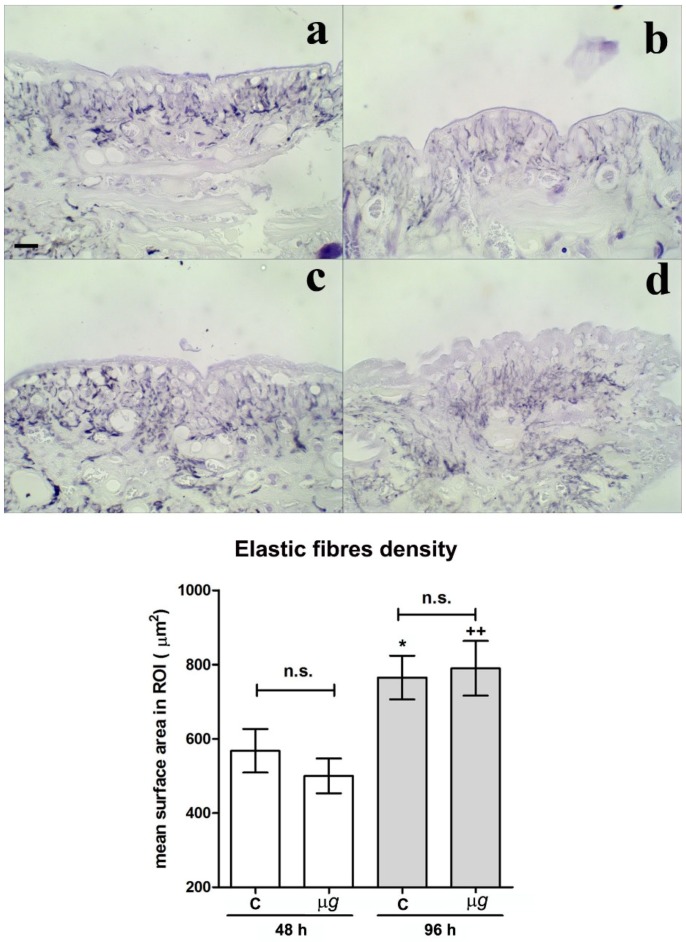
Effect of modeled *µg* on in vivo model of sutured wound healing (*Hirudo medicinalis*): elastic fibre content in the connective tissue at the wound site. Light microscopical features and morphometric quantitation of paraldehyde fuchsin-stained elastic fibres at the wound site after 48 h and 96 h exposure to modeled *µg* (**b**,**d**), compared with 1× *g* controls (**a**,**c**). In the graph C *=* 1× *g*; bar = 20 µm; * *p* < 0.05 vs. C-48h, ++ *p* < 0.01 vs. *µg*-48 h, (*n* = 3); n.s. = not significant.

**Figure 4 ijms-21-00407-f004:**
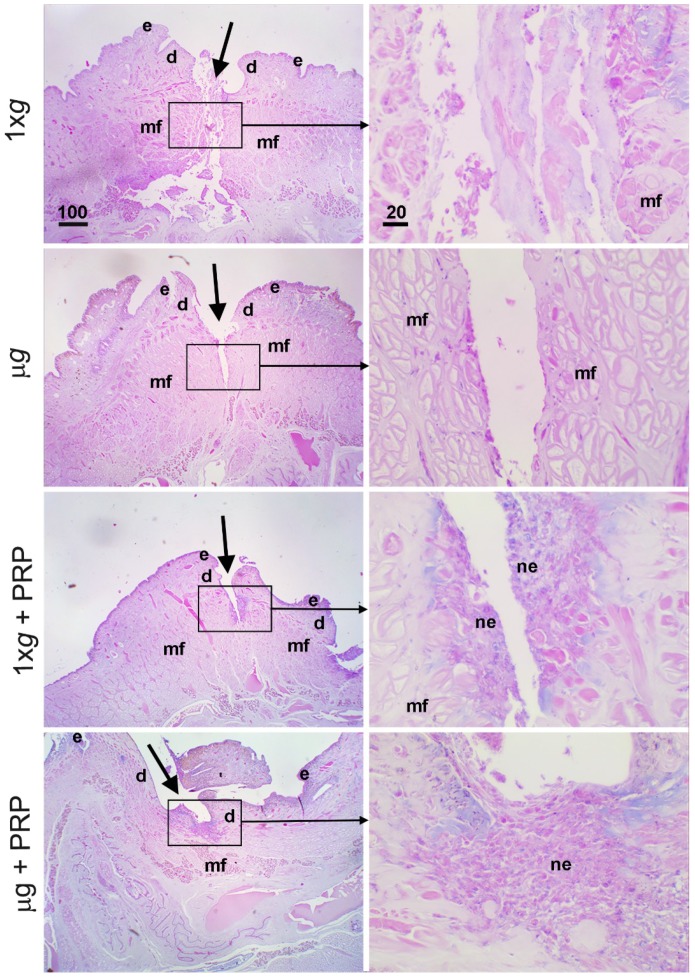
Effect of simulated *µg* and PRP treatment on in vivo model of sutured wound healing (*Hirudo medicinalis*): healing evolution at 144 h. Hematoxylin–eosin-stained cross sections. The panels show the state of the wounds (arrows) on the back of control animals (1× *g*) and animals exposed to simulated *µg* for 144 h, with or without PRP addition. Letterings are as follows: e, epidermis; d, dermis; mf, muscle fibers; ne, neo-epidermis. Calibration bars are in µm, as indicated.

**Figure 5 ijms-21-00407-f005:**
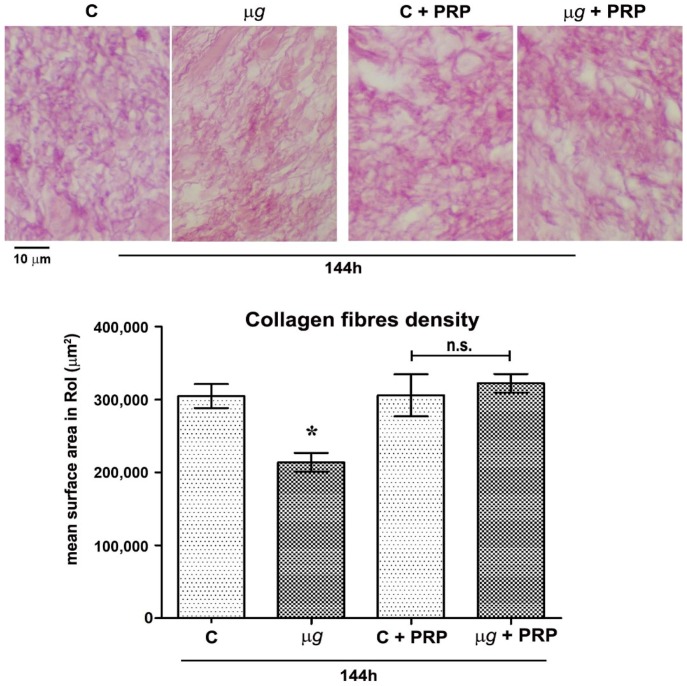
Effect of simulated *µg* and PRP treatment on in vivo model of sutured wound healing (*Hirudo medicinalis*): Collagen fibre content after 144 h. Light microscopical analysis and morphometric quantitation of picrosirius red-stained collagen fibres in the connective tissue at the wound site in control and *µg*-exposed animals, treated/untreated with PRP. C = 1× *g*; * *p* < 0.05 vs. all the other groups (*n* = 3); n.s. = not significant. To note the significantly higher collagen fiber density in *µg*-exposed animals treated with PRP compared with the untreated.

**Figure 6 ijms-21-00407-f006:**
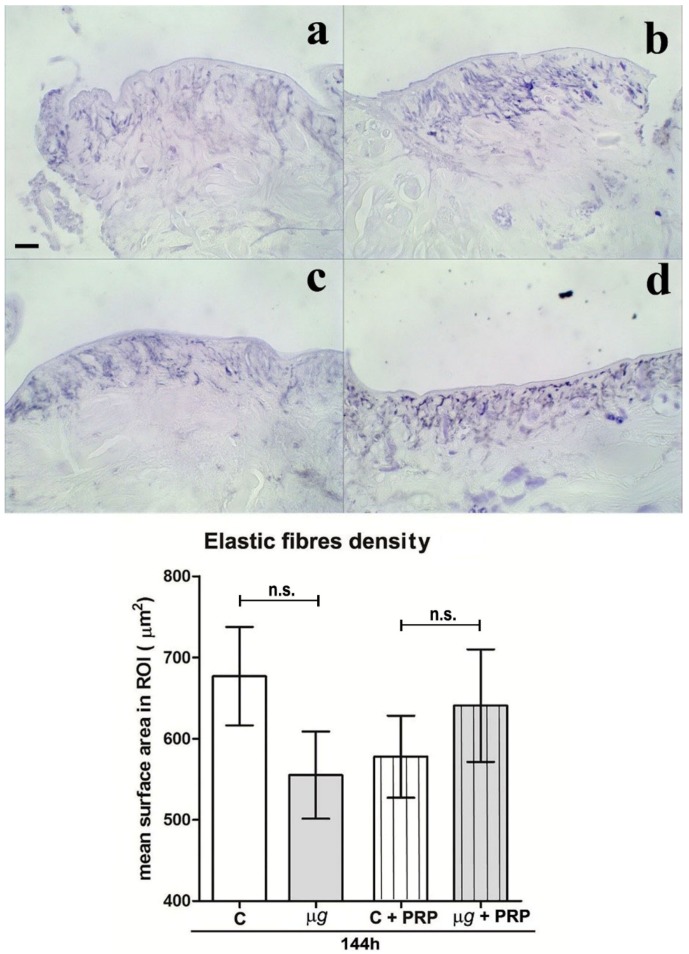
Effect of simulated *µg* and PRP treatment on in vivo model of sutured wound healing (*Hirudo medicinalis*): elastic fibre content after 144 h. Light microscopical analysis and morphometric quantitation of paraldehyde fuchsin-stained elastic fibres. 1× *g* (**a**); *µg* (**b**); 1× *g* + PRP (**c**); *µg* + PRP (**d**). Bar = 20 µm. Graph: C = 1× *g*, (*n* = 3); n.s. = not significant.

**Figure 7 ijms-21-00407-f007:**
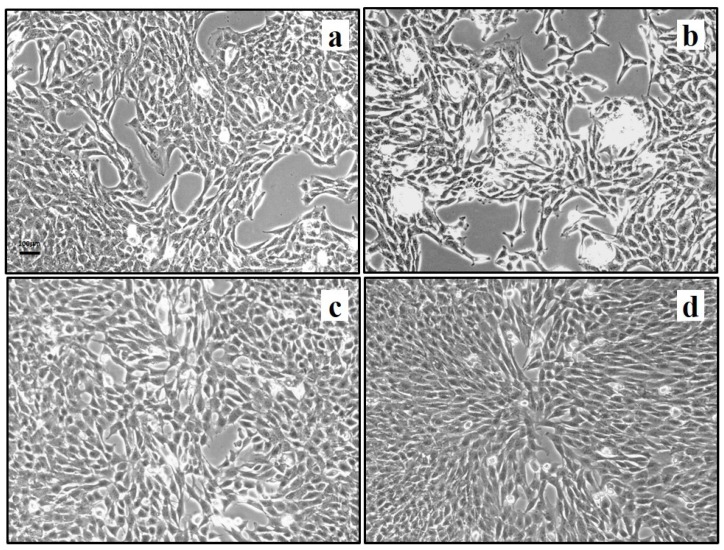
Effect of PRP treatment on NIH-3T3 cells exposed to modeled *µg*: cell morphology. Cells were exposed to modeled *µg* or maintained at 1× *g* (controls) for 72 h in presence/absence of PRP (1:1000). 1× *g* (**a**); *µg* (**b**); 1× *g* + PRP (**c**); *µg* + PRP (**d**). Bar = 100 µm.

**Figure 8 ijms-21-00407-f008:**
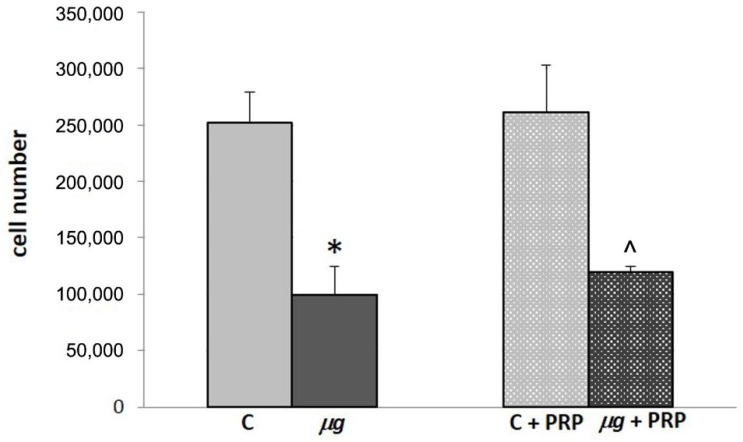
Effect of PRP treatment on NIH-3T3 cells exposed to modeled *µg*: cell count. Cells were exposed to modeled *µg* or maintained at 1× *g* (controls = C) for 72 h in the presence/absence of PRP (1:1000). At the end of exposure, cells were collected and counted with a Burker’s chamber. * *p* < 0.05 vs. C, ^ *p* < 0.05 vs. C + PRP, (*n* = 3).

**Figure 9 ijms-21-00407-f009:**
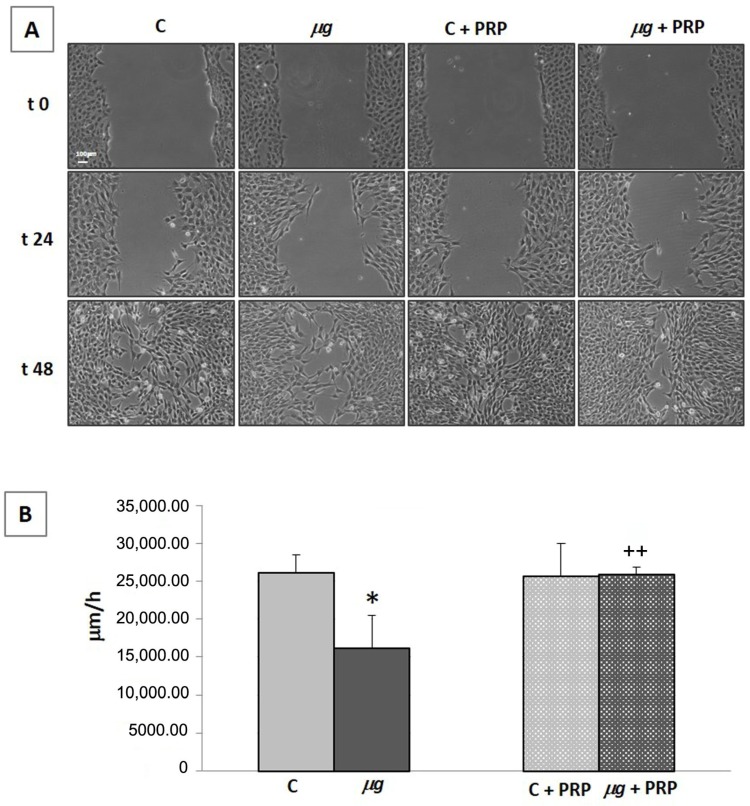
Effect of PRP treatment on NIH-3T3 cells exposed to modeled *µg*: cell migration—scratch assay. The scratch assay was performed during 72 h exposure to modeled *µg*, with or without PRP addition (1:1000). C = 1× *g*. Photos of the scratch at different times (**A**); migratory speed expressed as µm/h (**B**). Bar = 100 µm. * *p* < 0.05 vs. C, ++ *p* < 0.05 vs. µg, (*n* = 3).

**Figure 10 ijms-21-00407-f010:**
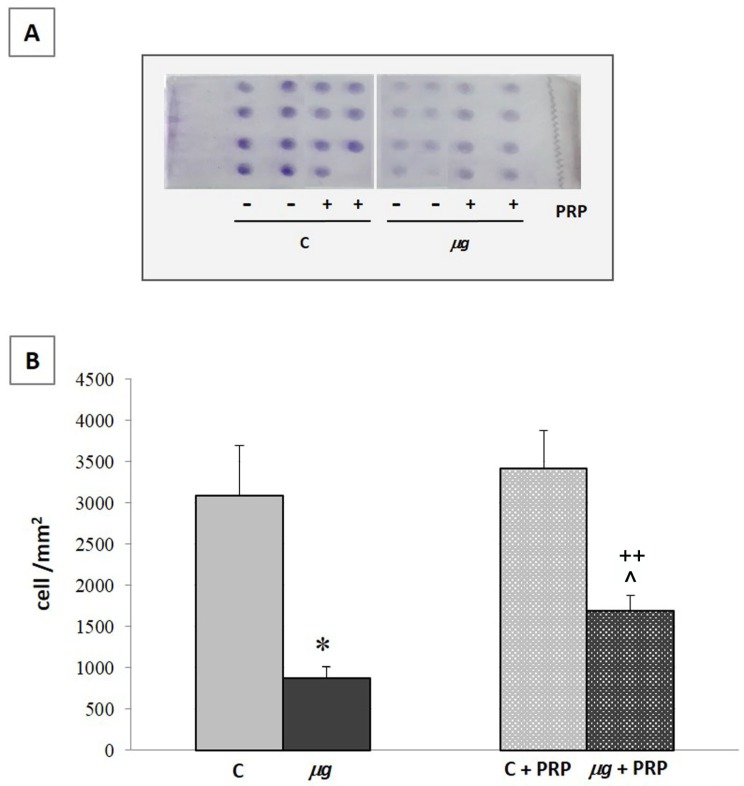
Effect of PRP treatment on NIH-3T3 cells exposed to modeled *µg*: cell migration—Boyden chamber assay. The Boyden chamber assay was performed immediately after 72 h exposure to modeled *µg*, with or without PRP addition (1:1000). C = 1× *g*. Representative images of the assay (**A**); number of migrated cells (**B**). * *p* < 0.05 vs. C, ++ *p* < 0.05 vs. µg, ^ *p* < 0.05 vs. C + PRP, (*n* = 3).

**Figure 11 ijms-21-00407-f011:**
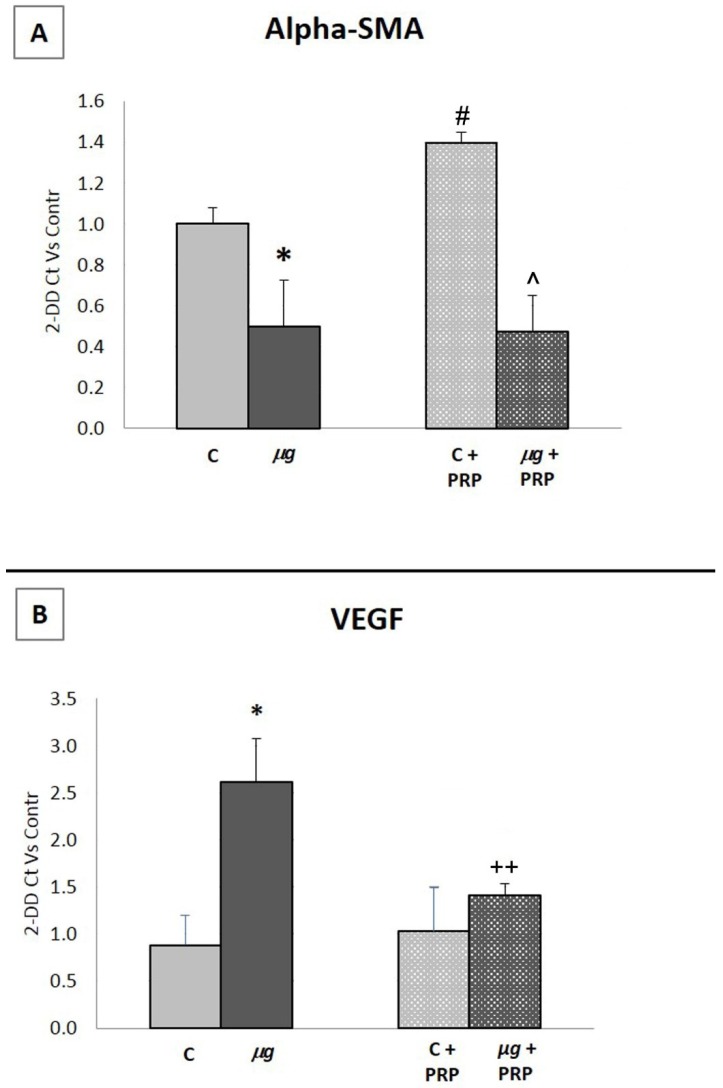
Effect of PRP treatment on α-smooth muscle actin (α-SMA) and vascular endothelial growth factor (VEGF) expression in NIH-3T3 cells exposed to modeled *µg*. Expression of α-SMA (**A**) and VEGF (**B**) assessed by real time PCR on total RNA from 1× *g* (controls) and *µg*-exposed cell samples treated/untreated with PRP during the 72 h exposure. C = 1× *g*. Data are expressed as 2^-ΔΔCt^ vs 1× *g* controls. (**A**) * *p* < 0.05 vs. C, # *p* < 0.05 vs. C, ^ *p* < 0.05 vs. C + PRP; (**B**) * *p* < 0.05 vs. C, ++ *p* < 0.05 vs. µg, (*n* = 3).

**Figure 12 ijms-21-00407-f012:**
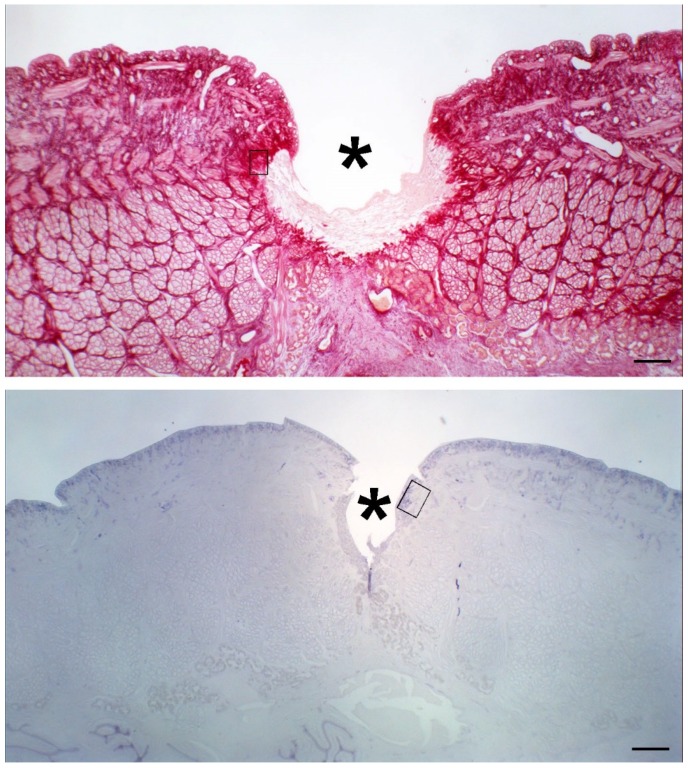
Representative transverse sections of leeches stained with picrosirius red (top) and paraldehyde fuchsin (bottom); the boxes show the tissue areas close to the surgical wound (asterisk) corresponding to the regions of interest (RoI) selected for collagen and elastic fibre densitometry, respectively. Bar = 200 μm.

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
