# Peer review of "Effect of Unloading Condition on the Healing Process and Effectiveness of Platelet Rich Plasma as a Countermeasure: Study on In Vivo and In Vitro Wound Healing Models"

_ijms, 2020, doi:10.3390/ijms21020407_

Round 1

Reviewer 1 Report

Cialdai et al. used a new in vivo sutured wound healing model in the leech under unloading conditions. They could confirm that microgravity caused a healing delay in their model and provided data on structural changes upon microgravity induction. Moreover, they used a microgravity-induced cell culture system of fibroblasts to analyze fibroblast proliferation and migration. In both systems the effect of platelet rich plasma on wound healing and fibroblast migration was analyzed and the authors argued that PRP prevented the effects on collagen, elastic fibres as well as fibroblast migration induced by microgravity.

This is an interesting study and I have some suggestions that may help to improve the manuscript.

There are many pictures, some of them may be combined. Labels within the pictures would increase understanding (treatment, time), as in Fig. 10 (in all others there is no labeling of the micrographs) In the Abstract the statement „PRP protective action is mostly due to its ability to effectively prevent the microgravity-induced impairment in fibroblast migration to the wound site“ appears overstated: although the authors showed that fibroblast migration was altered, they did not provide any data on other cell types involved in wound healing or on the in vivo wound bed. The authors should either provide additional data on other cell types or rephrase the Abstract. Some more background information of PRP and its (known) benefits on wound healing might help a non-experienced reader. The Results section might profit from some information on WHY the specific experiments were performed. Introductory sentences like „to determine whether xxx, yyy was performed“ might be of help. In the in vivo wound healing model the authors analyzed the time points 48h, 96h and 144h. The authors may want to mention the wound healing phase this time points would correspond to. Why was the time point 144h selected for PRP treatment? In addition, for non-experts it is difficult to see the wound area in the provided micrographs. Why was a different area chosen for analysis of histology, collagen and elastic fibre content? The authors should mark/highlight the different areas of the wound in the leech (granulation tissue, neo-epidermis, dermis and epidermis from healthy skin) and clearly indicate which areas are displayed/analyzed for the different parts of the study. The data on histological/structural changes within the wound are quite descriptive. Can the authors provide some kind of quantification in addition to the representative pictures to support their conclusions (Fig. 1, 2, 5; wider wounds, loose epithelium, less organized epithelium, PRP promoted wound healing by narrowing the surgical wound and enhancing re-epithelialization, …)? Figure 5d looks so much different from the other pictures (mesh-like structures, that are not seen in Fig. 1 and 2). Is it a specific induction by microgravity at 144h? The authors may comment on that. The authors found that microgravity induced cell aggregate formation in NIH3T3 cultures and argued that cell proliferation was significantly impaired. However, the authors only analyzed the cell count of their cultures. The authors need to determine whether cell aggregation and reduced cell numbers were induced by cell death/apoptosis or indeed by reduced proliferation rate (e.g. cell cycle analysis).      The data on aSMA and VEGF gene expression is again rather descriptive. The authors may explain why those two marker genes were used, what their regulation means, …

Author Response

REVIEWER 1

Authors

There are many pictures, some of them may be combined. Labels within the pictures would increase understanding (treatment, time), as in Fig. 10 (in all others there is no labeling of the micrographs).

We combined Figs 1 and 2. We labeled the micrographs as suggested by the referee

In the Abstract the statement „PRP protective action is mostly due to its ability to effectively prevent the microgravity-induced impairment in fibroblast migration to the wound site“ appears overstated: although the authors showed that fibroblast migration was altered, they did not provide any data on other cell types involved in wound healing or on the in vivo wound bed. The authors should either provide additional data on other cell types or rephrase the Abstract.

The abstract has been rephrased.

Some more background information of PRP and its (known) benefits on wound healing might help a non-experienced reader.

Background information on PRP has been added.

The Results section might profit from some information on WHY the specific experiments were performed. Introductory sentences like „to determine whether xxx, yyy was performed“ might be of help.

Introductory sentences and further information have been added.

In the in vivo wound healing model the authors analyzed the time points 48h, 96h and 144h. The authors may want to mention the wound healing phase this time points would correspond to.

Information on the healing phase and events has been added.

Why was the time point 144h selected for PRP treatment?

As explained in “materials and methods”, paragraph 4.2 “Exposure to modeled µg – Leeches”, after144h exposure, the effects of µg on the healing process were stronger than those observed at shorter exposure times. For this reason, 144h exposure was selected to evaluate the effectiveness of PRP as a countermeasure.

In addition, for non-experts it is difficult to see the wound area in the provided micrographs.

Arrows have been added to figure 4 (current version; in was figure 5 in the original version) pointing at the wounded dorsal surface of the leechs.

Why was a different area chosen for analysis of histology, collagen and elastic fibre content?

Indeed, except for the obvious differences in the histological slides stained for routine histology, collagen and elastic fibres (which anyhow were prepared using consecutive sections), similar tissue areas were chosen for the different stains: in particular, the regions of interest (RoI) used for collagen and elastic fibre morphometry were selected from the sub-epidermal and inter-muscular stroma flanking the margins and bed of the wound

The authors should mark/highlight the different areas of the wound in the leech (granulation tissue, neo-epidermis, dermis and epidermis from healthy skin) and clearly indicate which areas are displayed/analyzed for the different parts of the study.

Proper lettering was added to the panels in Figure 4 (in the original version it was Figure 5), as requested.

The data on histological/structural changes within the wound are quite descriptive. Can the authors provide some kind of quantification in addition to the representative pictures to support their conclusions (Fig. 1, 2, 5; wider wounds, loose epithelium, less organized epithelium, PRP promoted wound healing by narrowing the surgical wound and enhancing re-epithelialization, …)?

Collagen and elastic fibre density in the tissue has been quantified.

Quantification of wound width / depth in the H&E-stained slides is very difficult to perform in objective, unbiased manner. Leeches are annelid. They have an evident metamerism of the whole body, which appears divided into many rings. Therefore, the surface of the body presents undulations and folds. Accurate wound width / depth measurements would be difficult to perform and to compare. For this reason, when the leech is used as a model to study wound healing mechanisms, wound width / depth measurements are not reported (see also quoted papers of other authors). This fact could be considered a disadvantage in the use of the leech model. However, the leech model is considered particularly suitable for the study of some mechanisms underlying wound healing, such as fibroblast function and ECM synthesis and remodeling, that result extremely similar to those of mammals.

For consistency, we have modified the text by removing any reference to the length and width of the wounds.

Figure 5d (Fig. 4 in the revised version) looks so much different from the other pictures (mesh-like structures, that are not seen in Fig. 1 and 2). Is it a specific induction by microgravity at 144h? The authors may comment on that.

The peculiar mesh-like structures visible in the panel (Fig 4 in the revised version) actually are longitudinally sectioned muscle fibres. These can also be seen at the margins of other panels b (right) and f (left). Likely due to the very scarce de novo formation of reparative connective tissue in the leeches subjected to microgravity, the dorsal muscle tissue is still adjacent to the wound margins despite the considerable time elapsed from surgery.

The authors found that microgravity induced cell aggregate formation in NIH3T3 cultures and argued that cell proliferation was significantly impaired. However, the authors only analyzed the cell count of their cultures. The authors need to determine whether cell aggregation and reduced cell numbers were induced by cell death/apoptosis or indeed by reduced proliferation rate (e.g. cell cycle analysis).   

The three-dimensional growth observed after microgravity exposure is a known consequence of unloading conditions on different cell types. A comment on this topic has been added in the discussion.

As regards the decrease in proliferation, it has already been published that microgravity affects cell proliferation [Aleshcheva G et al., 2013; Kruger et al., 2019] in different cell types. Sun et al. reported that microgravity-induced reduction in osteoblast proliferation is due to an arrest in cell cycle progression [Sun et al., 2019]. In agreement with Sun, in preliminary experiments on fibroblasts cultured in microgravity modeled by a Rotating Cell Culture System (RCCS) we found a temporary arrest of the cell cycle, with repercussion on proliferation. These data are supported by the absence of necrosis and/or apoptotic bodies in all the images of the cell cultures (see Fig. 7 in the revised version, corresponding to Fig. 8 in the original version of the paper). It is known that microgravity induces an increase in apoptosis in some cell population but, to our knowledge, it has never been described in fibroblasts. We ourselves, studying fibroblasts in simulated microgravity with both RCCS (Cialdai et al., 2017) and RPM, have not found signs of apoptosis but a temporary arrest of the cell cycle with increase of the cells in phase “S”. This fact lead to a temporary decrease in proliferation followed by a slight increase: when the cell cycle restarts, the cells are partially synchronized. We have not shown these data because they have been observed modeling microgravity with RCCS, while the data presented in this study have been obtained using RPM.

The data on αSMA and VEGF gene expression is again rather descriptive. The authors may explain why those two marker genes were used, what their regulation means, …

Data on αSMA and VEGF expression are expressed quantitatively. We thank the reviewer for the suggestion to better explain the choice to analyze these markers. These topics have been specified in “Results” and “Discussion”.

Reviewer 2 Report

Effect of unloading condition on the healing process and effectiveness of platelet rich plasma as a countermeasure: study on in vivo and in vitro wound healing models is interesting and important as it introduced the leech model (as well utilizing the cultured cell scratch model), and has important negative findings as well as the positive findings.

The introduction could use some work as the quoted articles on "Studies on animal models have given controversial results and no definite conclusions, but most of them evidenced delayed and defective healing [13-15]" are two articles on mechanical unloading and a brief review. I think it is important to state that the data is from unloading studies. This article is about using platelet rich plasma to enhance wound healing during random positioning device assay of wounded leaches and cell culture. he introduction should be focused on these issues, not a general review (of which there are many).

The methods are detailed and appropriate. Furhter details of the exact rPM setting would be helpful (please see The Bonn Criteria: Minimal Experimental Parameter Reporting for Clinostat and Random Positioning Machine Experiments with Cells and Tissues. Microgravity Science and Technology February 2011, Volume 23, Issue 2, pp 271–275).

The histology is relevant and impressive, but entirely non quantitative and only shows selected images. The discussion of the histology should acknowledge the setting and lack of quantitation: the setting is important for data interpretation.

The assays of wound parameters are informative. n=3 in some diagrams and the large standard deviations due to lack of more replicates and disappointing as they dilute the significance of the study.

Overall an interesting study which moves the field forward. Better presentation of context and relevance would greatly strengthen the study.

Author Response

REVIEWER 2

Author

The introduction could use some work as the quoted articles on "Studies on animal models have given controversial results and no definite conclusions, but most of them evidenced delayed and defective healing [13-15]" are two articles on mechanical unloading and a brief review. I think it is important to state that the data is from unloading studies. This article is about using platelet rich plasma to enhance wound healing during random positioning device assay of wounded leaches and cell culture. he introduction should be focused on these issues, not a general review (of which there are many).

We added to the introduction a paragraph focused on PRP and its ability to enhance wound healing. We maintained the “general review part”, because we think it is useful (in particular for readers not-expert in space biology) to explain the problems related to wound healing in microgravity and the need for countermeasures. We added further quotations [13-18] to report data obtained both in real and modeled unloading conditions. 

The methods are detailed and appropriate. Furhter details of the exact RPM setting would be helpful (please see The Bonn Criteria: Minimal Experimental Parameter Reporting for Clinostat and Random Positioning Machine Experiments with Cells and Tissues. Microgravity Science and Technology February 2011, Volume 23, Issue 2, pp 271–275).

Information on RPM setting in the section “material and methods” has been improved following The Bonn Criteria: Minimal Experimental Parameter Reporting for Clinostat and Random Positioning Machine Experiments with Cells and Tissues. Microgravity Science and Technology February 2011, Volume 23, Issue 2, pp 271–275.

The histology is relevant and impressive, but entirely non quantitative and only shows selected images. The discussion of the histology should acknowledge the setting and lack of quantitation: the setting is important for data interpretation.

Collagen and elastic fibre density in the tissue has been quantified.

Quantification of wound width / depth in the H&E-stained slides is very difficult to perform in objective, unbiased manner. Leeches are annelid. They have an evident metamerism of the whole body, which appears divided into many rings. Therefore, the surface of the body presents undulations and folds. Accurate wound width / depth measurements would be difficult to perform and to compare. For this reason, when the leech is used as a model to study wound healing mechanisms, wound width / depth measurements are not reported (see also quoted papers of other authors). This fact can be considered a disadvantage in the use of the leech model. However, the leech model is considered particularly suitable for the study of some mechanisms underlying wound healing, such as fibroblast function and ECM synthesis and remodeling, that result extremely similar to those of mammals.

For consistency, we have modified the text by removing any reference to the length and width of the wounds.

The assays of wound parameters are informative. n=3 in some diagrams and the large standard deviations due to lack of more replicates and disappointing as they dilute the significance of the study.

The reviewer is right: n=3 is the minimum for a reliable statistical analysis and, possibly, some of the reported differences among the experimental groups are under-estimated. However the wounded leeches must be placed one for each flask, because they attack each other. In order to model microgravity correctly, we preferred to put a limited number of flasks on the center of the RPM.

However, the experiments have been repeated three times.

Round 2

Reviewer 1 Report

The authors addressed most of my comments satisfactorily but I have still few comments.

The authors nicely displayed the region of interest for the histological analysis, but in the manuscript it is still not clear which areas are displayed/analyzed for collagen and elastic fibre content. They explained in their reply that "the regions of interest (RoI) used for collagen and elastic fibre morphometry were selected from the sub-epidermal and inter-muscular stroma flanking the margins and bed of the wound" and need to specify this in the manuscript (optimally by providing an overview of the whole section and indicating the area, similar to current Fig. 4).

The authors need to comment on the "very scarce de novo formation of reparative connective tissue in the leeches subjected to microgravity..." in the manuscript (Fig. 4), not only in their reply to the reviewer.

The authors commented on the three-dimensional growth but failed to provide any data on proliferation. They stated in their response that by "studying fibroblasts in simulated microgravity with both RCCS (Cialdai et al., 2017) and RPM, [they] have not found signs of apoptosis but a temporary arrest of the cell cycle with increase of the cells in phase “S”." They need to include such data to show that indeed in their system the proliferation was affected. Otherwise they should not state that "NIH-3T3 proliferation was evaluated" or "NIH-3T3 proliferation was significantly impaired", although they might speculate on such an effect based on their cell count analysis. Then they need to rewrite the corresponding results and discussion sections. 

Author Response

REVIEWER 1

Authors

The authors addressed most of my comments satisfactorily but I have still few comments.

The authors nicely displayed the region of interest for the histological analysis, but in the manuscript it is still not clear which areas are displayed/analyzed for collagen and elastic fibre content. They explained in their reply that "the regions of interest (RoI) used for collagen and elastic fibre morphometry were selected from the sub-epidermal and inter-muscular stroma flanking the margins and bed of the wound" and need to specify this in the manuscript (optimally by providing an overview of the whole section and indicating the area, similar to current Fig. 4).

A figure of representative picrosirius red- and paraldehyde fuchsin-stained transverse sections of leeches has been added to show tissue areas in which the regions of interest (RoI) were selected for - respectively - collagen and elastic fibre densitometry.

The authors need to comment on the "very scarce de novo formation of reparative connective tissue in the leeches subjected to microgravity..." in the manuscript (Fig. 4), not only in their reply to the reviewer.

We added the comment in the manuscript (Results, 2.1.1.-Histological analysis)

The authors commented on the three-dimensional growth but failed to provide any data on proliferation. They stated in their response that by "studying fibroblasts in simulated microgravity with both RCCS (Cialdai et al., 2017) and RPM, [they] have not found signs of apoptosis but a temporary arrest of the cell cycle with increase of the cells in phase “S”." They need to include such data to show that indeed in their system the proliferation was affected. Otherwise they should not state that "NIH-3T3 proliferation was evaluated" or "NIH-3T3 proliferation was significantly impaired", although they might speculate on such an effect based on their cell count analysis. Then they need to rewrite the corresponding results and discussion sections. 

As regards proliferation, we rewrote the corresponding Results and Discussion.

We preferred to mention our findings on the cell cycle arrest in microgravity, together with results from other authors, in order to explain our speculation on the possible cause of a decrease in fibroblast number, rather than include the above findings in this manuscript as part of the study, because they were obtained using a RCCS to model microgravity, while in the study here described an RPM was used. Therefore, although both facilities are widely accepted to simulate microgravity, the experimental conditions are different.

Reviewer 2 Report

Thank you for greatly improving the manuscript according to the suggestions: it should help the reviewers a lot.

I have one caveat. It might be worth mentioning the currently FDA approached leech therapy and the indications for its use.

Author Response

REVIEWER 2

Authors

Thank you for greatly improving the manuscript according to the suggestions: it should help the reviewers a lot.

I have one caveat. It might be worth mentioning the currently FDA approached leech therapy and the indications for its use.

We added information in the manuscript (Materials and Methods, 4.1-Animals and Surgical Treatment)